# Artifact-Free Microstructures in the Interfacial Reaction between Eutectic In-48Sn and Cu Using Ion Milling

**DOI:** 10.3390/ma16093290

**Published:** 2023-04-22

**Authors:** Fu-Ling Chang, Yu-Hsin Lin, Han-Tang Hung, Chen-Wei Kao, C. R. Kao

**Affiliations:** Department of Materials Science and Engineering, National Taiwan University, Taipei 106216, Taiwan

**Keywords:** low-temperature soldering, interconnection, Cu-In-Sn intermetallic compounds, interfacial reaction, mechanical properties

## Abstract

Eutectic In-48Sn was considered a promising candidate for low-temperature solder due to its low melting point and excellent mechanical properties. Both Cu_2_(In,Sn) and Cu(In,Sn)_2_ formation were observed at the In-48Sn/Cu interface after 160 °C soldering. However, traditional mechanical polishing produces many defects at the In-48Sn/Cu interface, which may affect the accuracy of interfacial reaction investigations. In this study, cryogenic broad Ar^+^ beam ion milling was used to investigate the interfacial reaction between In-48Sn and Cu during soldering. The phase Cu_6_(Sn,In)_5_ was confirmed as the only intermetallic compound formed during 150 °C soldering, while Cu(In,Sn)_2_ formation was proven to be caused by room-temperature aging after soldering. Both the Cu_6_(Sn,In)_5_ and Cu(In,Sn)_2_ phases were confirmed by EPMA quantitative analysis and TEM selected area electron diffraction. The microstructure evolution and growth mechanism of Cu_6_(Sn,In)_5_ during soldering were proposed. In addition, the Young’s modulus and hardness of Cu_6_(Sn,In)_5_ were determined to be 119.04 ± 3.94 GPa and 6.28 ± 0.13 GPa, respectively, suggesting that the doping of In in Cu_6_(Sn,In)_5_ has almost no effect on Young’s modulus and hardness.

## 1. Introduction

Due to the toxicity of Pb, there has been increasing demand for the use of Pb-free solders since the beginning of the 21st century. Currently, lead-free Sn-based solder alloys are widely used in electronic packaging systems [1,2]. Although using Sn as a soldering material avoids Pb toxicity, it results in an increase in soldering temperature, as the melting point of most Sn-based solders is approximately 200 °C. Temperature-sensitive chips are unable to withstand this high-temperature bonding, so low-temperature soldering materials need to be developed. A reduction in soldering temperature could also decrease the energy used in the soldering process, which would be environmentally beneficial.

Eutectic alloys are ideal because of their low soldering temperatures, making them easy to apply in industrial production. Therefore, Sn-based binary alloys doped with Ga, Zn, Bi, and In have been recognized as potential candidates for low-temperature bonding [1,2,3]. However, these alloying elements have drawbacks; for instance, using an alloy containing Ga is impractical due to its low melting point. The addition of Zn in the alloy results in poor wettability, reliability, and corrosion resistance, and the addition of Bi can cause brittle failure and segregation in the alloy. The eutectic In-48Sn alloy is a promising soldering material, as it is non-toxic and has a melting point of 120 °C. This alloy also has good wettability, excellent ductility, and a long fatigue life [4]. In addition, it was reported that various diameters of In-48Sn solder wires and rods can be produced easily because of the ductility and high elongation of In-48Sn [5]. For these reasons, eutectic In-48Sn alloy can be used in temperature-sensitive chips, such as biochips and flexible chips. 

Solid–liquid interdiffusion (SLID) bonding is a well-developed and widely used technique in the electronic packaging industry. Generally, a SLID metal system consists of two layers: a high-melting-point substrate layer and a low-melting-temperature solder layer. During the bonding process, the joint is heated to a temperature greater than the melting point of the solder layer. The molten solder reacts with the substrate layer to form at least one IMC and continues until the entire solder layer transforms into IMCs with much higher melting temperatures than the original solder. Compared with the other conventional techniques used in the electronic packing industry, the most important benefit of SLID bonding is that the joint can be bonded at low temperatures and used in high-temperature applications due to the formation of high-melting-temperature IMCs. 

Several studies have investigated the interfacial reaction between In-48Sn solder and Cu substrate. Phase identification of the formed IMCs at the In-48Sn/Cu interface was performed; two IMCs, Cu_2_(In,Sn) and Cu(In,Sn)_2_, formed after reflowing at 160 °C [6].The voids existing at the In-48Sn/Cu interface were proven to be Kirkendall voids formed during reflowing and the solid-state aging process [7,8]. The growth mechanism of Cu_2_(In,Sn) and Cu(In,Sn)_2_ and the phase transformation between these two phases were investigated [9,10,11,12,13,14,15]. It was found that Cu(In,Sn)_2_ transformed to Cu_2_(In,Sn) when aging above 60 °C, while Cu_2_(In,Sn) transformed to Cu(In,Sn)_2_ when aging below 60 °C. The relationship between IMC thickness and aging time was also determined. In an investigation of Cu addition to In-48Sn solder, the observed Cu-rich IMC was reported to be η-Cu_6_(Sn,In)_5_ [16,17]. In the Cu/In-48Sn/Cu interfacial reaction, where the In-48Sn is completely consumed, Cu_6_(Sn,In)_5_ and Cu_3_(Sn,In) were observed at the interface after 250 °C bonding; only Cu_6_(Sn,In)_5_ was observed after 200 °C to 160 °C bonding [18,19,20,21,22]. The results of previous works imply that there is extremely fast formation of IMCs at the In-48Sn/Cu interface, which makes In-48Sn a promising SLID material [23]. However, the In-48Sn/Cu interface is easily destroyed during mechanical polishing, and the lack of an artifact-free In-48Sn/Cu interface may restrict the accurate analysis of microstructure evolution and phase identification at the interface. 

In this study, the interfacial reaction between molten In-48Sn and Cu at 150 °C from tens of seconds to tens of minutes was studied. Cryogenic ion milling was used to produce an artifact-free interface so that the interface microstructure evolution, phase identification, and IMC growth mechanism could be investigated more accurately. Phase identification of each phase was confirmed by both chemical composition and crystallographic structure. Additionally, the IMC formed at the interface caused by room-temperature storage after soldering was also clarified.

## 2. Experimental Method

### 2.1. Sample Preparation

The eutectic In-48Sn solder was prepared by mixing and melting high-purity In (99.99%) and Sn shots (99.99%) under a vacuum. Oxygen-free Cu plates were cut to a size of 10 mm × 10 mm × 1 mm, ground using silicon carbide sandpaper, and polished with 3 μm and 1 μm polycrystalline diamond polishing fluid.

As shown in Figure 1, the amount of solder used was 0.15 (±0.001) g, and the circular reaction area had a diameter of 3 mm. The copper substrates were cleaned with sulfuric acid and isopropanol solutions. Halogen-free flux and Kapton tape with a temperature resistance exceeding 150 °C were used for sample preparation.

Once the In-48Sn solder was added, the sample was placed in a convection oven at 150 (±1) °C. After the liquid–solid reaction was complete, the sample was quenched in water.

### 2.2. Cross-Section Observation and Ion Milling

The alloy sample was metallographically polished prior to cross-sectional observation. However, during mechanical polishing, the silicon carbide particles on the sandpaper and diamond abrasives in the polishing fluid were embedded in the soft In-Sn alloy. It has been reported that ion milling can be used in place of mechanical polishing for samples that are damaged by mechanical polishing [24]. An ion milling system (Hitachi, IM4000Plus, Tokyo, Japan) equipped with a liquid-nitrogen cryogenic system was used to prevent the In-48Sn solder in the sample from melting due to the heat generated during the ion milling process.

The sample was ion milled at −50 °C for 1.5 h to obtain an artifact-free cross-sectional micrograph of the solder/substrate interface, as shown in Figure 2. Scanning electron microscopy (SEM; Hitachi SU-5000, Tokyo, Japan) was used to observe the morphology and evolution of the solder/substrate interface. The chemical compositions of the solder and IMCs at the solder/substrate interface were analyzed using energy-dispersive X-ray spectroscopy (EDS; Bruker, Billerica, MA, USA) and an electron probe microanalyzer (EPMA; JEOL JXA-8530FPlus, Tokyo, Japan).

### 2.3. Top-View Observation and Etching

The top-view morphology and crystallographic structures of the IMCs were observed using SEM and X-ray diffraction (XRD), respectively. The alloy sample was chemically etched to remove the excess solder and expose the IMCs. This allowed us to observe the top-view morphology and identify the crystallographic structures of the IMCs. A high-power X-ray diffractometer (18 kW) (Rigaku, TTRAX3, Tokyo, Japan) with a Cu-Kα source was used to identify the crystallographic structures of the IMCs.

### 2.4. Mechanical Property Determination

Because the thickness of the IMCs was on the order of micrometers, nanoindentation testing (Hysitron TI 980 TriboIndenter with a Berkovich tip, Billerica, MA, USA) was used to analyze the mechanical properties of Cu_6_(Sn,In)_5_ in this study.

## 3. Results and Discussion

### 3.1. Interfacial Microstructure

The In-48Sn/Cu sample was soldered at 150 °C for 80 min, followed by water quenching. The cross-section of the interface is shown in Figure 3a, and the top-view morphology of the IMCs after all the In-48Sn solder was removed by the etching solution is shown in Figure 3b. As shown in Figure 3, there are two distinct microstructure regions at the interface. The region in contact with the solder has a large quantity of rod-type IMCs, while the region in contact with the substrate is composed of a solder/IMC mixture. Similar observations have been reported in the solid–liquid reaction in Cu/In binary systems [24]. There are differences in the interface shape and phase of the IMCs between the In-48Sn/Cu and Cu/In systems. According to EDS quantitative analysis, the elemental composition of the IMC in both microstructures can be described as 56Cu-18In-26Sn (at.%), indicating that it is most likely to be Cu_6_(Sn,In)_5_ rather than Cu_2_(In,Sn). This result will be confirmed using EPMA and XRD analysis, as discussed below.

### 3.2. Phase Identification

The EPMA elemental mapping and quantitative analysis are shown in Figure 4 and Table 1, respectively. The In-48Sn solder matrix was separated into *β* and γ phases, which is consistent with the reported In-Sn phase diagram [25]. As mentioned above, the elemental composition of the IMC in the mixed layer and the rod-type IMC is Cu_6_(Sn,In)_5_.

The compound Cu_6_Sn_5_ is a common IMC in Sn-containing solder joints and is well known for its polymorphic transformation from η-Cu_6_Sn_5_ (hexagonal, space group P63/mmc) to η’-Cu_6_Sn_5_ (monoclinic, space group C2/c). This polymorphic transformation is accompanied by a volume change, which induces the formation and propagation of cracks and is considered an important factor leading to packaging failure.

XRD analysis was used to confirm the crystallographic structure of the IMC at the interface; the obtained diffraction pattern is shown in Figure 5. The remaining solder was completely etched; therefore, all of the peaks in the XRD pattern were from the IMCs and substrate. As shown in Figure 5, only Cu_6_(Sn,In)_5_ and Cu peaks are present. This confirms that the chemical composition of both the rod-type IMCs and IMCs in the mixture layer is Cu_6_(Sn,In)_5_, which is consistent with the results of the EPMA quantitative analysis. The appearance of several weak peaks in the XRD pattern can help distinguish whether the appearance of the IMC belongs to the η-Cu_6_Sn_5_ or η’-Cu_6_Sn_5_ phase. Weak peaks would indicate monoclinic η’-Cu_6_Sn_5_, as it has relatively poor symmetry [12,23,24,26]. As shown in Figure 5, no monoclinic peaks can be observed from the XRD pattern, meaning that the Cu_6_(Sn,In)_5_ at the interface is the high-temperature phase of η-Cu_6_Sn_5_. This result is consistent with previous work, which found that the addition of In stabilized the high-temperature phase η-Cu_6_Sn_5_ at room temperature [26]. The results show that each diffraction peak from Cu_6_(Sn,In)_5_ slightly deviates from In-free η-Cu_6_Sn_5_. This is caused by a change in the interplanar spacing due to the replacement of Sn by In. The crystallographic structure information of Cu, η-Cu_6_Sn_5_, and η’-Cu_6_Sn_5_ is from the Joint Committee on Powder Diffraction Standards (JCPDS) data file numbers 04-0836, 02-0713, and 45-1488.

### 3.3. Microstructure Evolution

As mentioned previously, there are two different microstructures at the interface: rod-type Cu_6_(Sn,In)_5_ and a solder/Cu_6_(Sn,In)_5_ mixture layer. The cross-section microstructure evolution of the liquid–solid reaction after soldering at 150 °C for 0.5 min, 2 min, 10 min, 20 min, 40 min, and 80 min is shown in Figure 6a–f. As shown in Figure 6, the thickness of the solder/Cu_6_(Sn,In)_5_ layer formed during soldering increased with time. Furthermore, there is a non-uniform distribution in the solder/Cu_6_(Sn,In)_5_ mixture layer, especially during long-term soldering. This non-uniformity can be described as Cu_6_(Sn,In)_5_ dissolved in molten In-48Sn. While Sn and In diffuse downward to form Cu_6_(Sn,In)_5_, In-48Sn solder simultaneously dissolves Cu_6_(Sn,In)_5_. This can explain why the proportion of Cu_6_(Sn,In)_5_ decreases as it moves toward the In-48Sn solder. 

To investigate the formation of rod-type Cu_6_(Sn,In)_5_ at the interface, the sample after soldering at 150 °C for 80 min was observed with a low magnification SEM image, as shown in Figure 7. In addition to the rod-type Cu_6_(Sn,In)_5_ precipitates at the interface, water quenching results in some sporadic rod-type Cu_6_(Sn,In)_5_ precipitation in the In-48Sn solder matrix far away from the interface. A precipitate-free zone could be observed above the In-48Sn/Cu interface. These results indicate that rod-type Cu_6_(Sn,In)_5_ forms during the cooling process, which is caused by a decrease in Cu solubility in the In-48Sn solder after solidification. During solidification, excess Cu in the In-48Sn solder precipitates. Cu close to the interface is then able to move to the interface and form heterogeneous precipitates when water quenching is applied. On the other hand, Cu that is far from the interface is only able to homogeneously precipitate in situ when quenched with water. However, Cu near the interface is still able to form heterogeneous precipitation. This allows a precipitate-free zone between the heterogeneous precipitation and homogeneous precipitation regions to form.

To understand the relationship between Cu concentration and precipitation, the amount of solder was reduced to 0.075 g, while the reaction area was held constant. Just as was performed above, the sample was soldered at 150 °C for 80 min. Figure 8a,b show the top-view morphology, while Figure 8c,d show the interface cross-sections of the two samples prepared with different amounts of solder. According to the cross-sectional microstructure and the position of the Cu_6_(Sn,In)_5_/Cu interface, the total Cu consumption is consistent between the two samples. This is because the Cu concentration in In-48Sn is well below the saturation concentration, and, therefore, the dissolution rate and amount of Cu_6_(Sn,In)_5_ are related to the soldering time rather than Cu concentration. Because Cu dissolved in In-48Sn is equal to Cu consumption, and Cu exists as Cu_6_(Sn,In)_5_, these two samples have the same Cu consumption and amount of Cu_6_(Sn,In)_5_. This indicates that the total Cu in both amounts of In-48Sn solder is the same. The molten In-48Sn solidified as *β* and γ phases when cooled to room temperature (25 °C). Because Cu is almost completely insoluble in both the *β* and γ phases, the supersaturated Cu in the solder matrix precipitates in the form of Cu_6_(Sn,In)_5_ after cooling. Based on these observations, the Cu concentration in the sample using 0.075 g of solder was twice that of the sample using 0.15 g of solder. This increase in Cu concentration results in denser rod-type Cu_6_(Sn,In)_5_ heterogeneous precipitation at the interface (Figure 8d). Conversely, the sample using 0.15 g solder has more homogeneous precipitation in its solder matrix (Figure 8c). Furthermore, the sum of the heterogeneous and homogeneous precipitation remains the same between the two samples. 

A previous study reported that discontinuous Cu_6_(Sn,In)_5_ around rod-type Cu_6_(Sn,In)_5_ was caused by spalling. However, based on the proposed formation mechanism and random growth direction of Cu_6_(Sn,In)_5_ from the top view, the discontinuous Cu_6_(Sn,In)_5_ was determined to be not due to spalling but rather truncated rod-type Cu_6_(Sn,In)_5_.

A schematic of the proposed mechanism for the interfacial reaction between the liquid In-48Sn and solid Cu at 150 °C is illustrated in Figure 9. At the beginning of soldering, a thin Cu_6_(Sn,In)_5_ layer forms at the In-48Sn interface. Then, the Cu_6_(Sn,In)_5_ layer continues to grow and is simultaneously dissolved by the molten In-48Sn solder as soldering proceeds. Finally, rod-type Cu_6_(Sn,In)_5_ precipitates after cooling due to the difference in Cu solubility between liquid and solid In-48Sn.

### 3.4. Room-Temperature Aging

A previous study reported the formation of both Cu_6_(Sn,In)_5_ and Cu(In,Sn)_2_ IMCs during soldering at 160 °C. However, the formation of Cu(In,Sn)_2_ at the interface after soldering at 150 °C was not observed in this study. It is possible that Cu(In,Sn)_2_ was not formed during soldering but during room-temperature storage. Because 25 °C is approximately 0.7 times the melting temperature of In, it has been shown to have an apparent creep at room temperature [27,28]. The diffusion of In at room temperature is relatively fast; therefore, interfacial reactions may take place, even at room temperature. To determine the effect of room-temperature storage, the as-bonded samples were stored at room temperature for different amounts of time. The cross-section of the interface after room-temperature storage for 24, 500, and 5000 h is shown in Figure 10a–c. In the sample aged for 24 h, only the Cu_6_(Sn,In)_5_ continuous layer and rod-type Cu_6_(Sn,In)_5_ are present at the interface. However, there is no longer rod-type Cu_6_(Sn,In)_5_ in the sample stored for 500 h. In contrast, a Cu(In,Sn)_2_ continuous layer formed between the solder matrix and the Cu_6_(Sn,In)_5_ layer. After 5000 h of storage at room temperature, the thickness of the Cu(In,Sn)_2_ continuous layer increases, and Cu_6_(Sn,In)_5_ becomes a discontinuous layer. This experiment demonstrates that the Cu(In,Sn)_2_ continuous layer formed during room-temperature storage after soldering, which was first reported in the Cu-In-Sn system. This storage process causes unintentional IMC growth under room temperature; hence, this phenomenon is called room-temperature aging.

Phase identification of the Cu(In,Sn)_2_ phase was determined using EPMA and TEM selected area electron diffraction (SAED). The EPMA elemental mapping of the sample aged for 500 h at room temperature is shown in Figure 11. As shown in the EPMA mapping, Cu(In,Sn)_2_ not only exists between the solder and Cu_6_(Sn,In)_5_ layer, but all the Cu_6_(Sn,In)_5_ precipitated in the solder matrix was transformed into Cu(In,Sn)_2_. The results of the EPMA quantitative analysis are shown in Table 2. TEM/SAED was used to further confirm the crystal structure of this phase because the thicknesses of these two IMCs are close to the EPMA resolution limit. 

Figure 12a shows a TEM bright-field image of both Cu_6_(Sn,In)_5_ and Cu(In,Sn)_2_. The SAED patterns of the Cu(In,Sn)_2_ (**b**) and Cu_6_(Sn,In)_5_ (**c**) regions are shown in Figure 12b,c, respectively.

The results of the SAED patterns indicate that the IMC in Figure 12b is Cu(In,Sn)_2_ with a zone axis [010], and the IMC in Figure 12c is Cu_6_(Sn,In)_5_ with a zone axis [121]. The crystallographic structure information of η-Cu_6_Sn_5_ is from the JCPDS 02-0713, while CuIn_2_ is from the data of Gossla et al. [29].

### 3.5. Mechanical Properties of Cu_6_(Sn,In)_5_

The mechanical properties of Cu_6_Sn_5_ have been widely studied; however, few studies have investigated the effect of In-doping on the mechanical properties of Cu_6_(Sn,In)_5_. The mechanical properties of the solder joint and the formed IMCs directly influence the reliability of the bonding results. Here, the effect of adding In on the mechanical properties of Cu_6_(Sn,In)_5_ was investigated.

The maximum applied load was 1200 μN, with a 240 μN/s loading rate, and the maximum load was held for 5 s. The load–displacement curve of Cu_6_(Sn,In)_5_ is shown in Figure 13. The Young’s modulus and hardness were averaged from the results of five different points. The measured Young’s modulus and hardness are listed in Table 3 alongside those reported for pure Cu_6_Sn_5_ [30]. As shown in Table 3, both the Young’s modulus and hardness of Cu_6_(Sn,In)_5_ are similar to those of pure Cu_6_Sn_5_.

## 4. Conclusions

In this study, the phase identification, microstructure evolution, and growth mechanism between molten In-48Sn and Cu substrates were investigated. The phase Cu_6_(Sn,In)_5_ was identified as the only IMC formed at the interface during soldering at 150 °C, which was confirmed by EPMA quantitative analysis and XRD. A solder/Cu_6_(Sn,In)_5_ mixture layer was formed during the soldering process, and the thickness of this layer increased with the soldering time. This layer formed through the simultaneous growth of the Cu_6_(Sn,In)_5_ layer and its dissolution by the molten In-48Sn solder. The amount of Cu_6_(Sn,In)_5_ in the layer decreases as it approaches the In-48Sn solder. Rod-type Cu_6_(Sn,In)_5_ formation was caused by the heterogeneous precipitation of supersaturated Cu in the In-48Sn solder due to a decrease in solubility after solidification.

The formation of Cu(In,Sn)_2_ at the interface was proven to be caused by room-temperature aging after soldering. This result indicates that the In-48Sn/Cu interface cannot be stored at room temperature and needs to be observed immediately after the experiment; otherwise, room-temperature aging will influence the results of the interfacial reaction. The Young’s modulus and hardness of Cu_6_(Sn,In)_5_ were determined to be 119.04 ± 3.94 GPa and 6.28 ± 0.13 GPa, respectively, indicating that the addition of In to Cu_6_(Sn,In)_5_ did not have a significant impact on Young’s modulus and hardness.

In this study, although the formation of Cu(In,Sn)_2_ caused by room-temperature aging was reported, the critical temperature at which Cu(In,Sn)_2_ starts to appear was not determined, so further studies are still needed.

## Figures and Tables

**Figure 1 materials-16-03290-f001:**
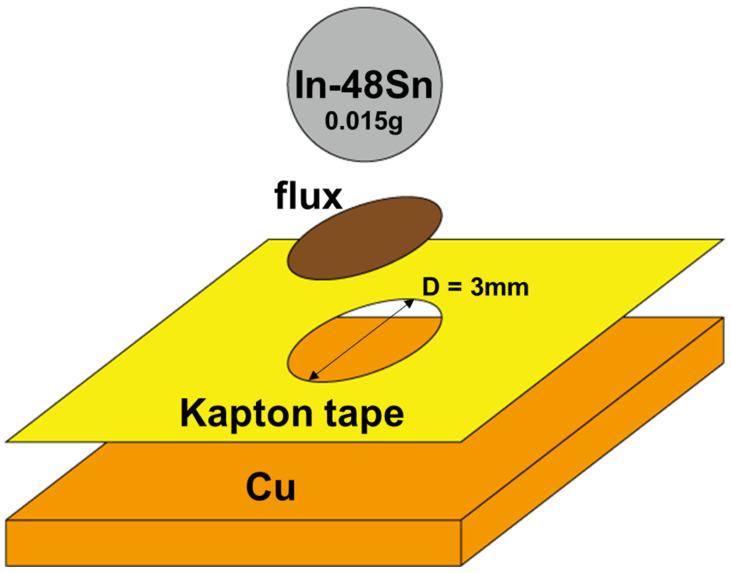
Schematic of the sample preparation.

**Figure 2 materials-16-03290-f002:**
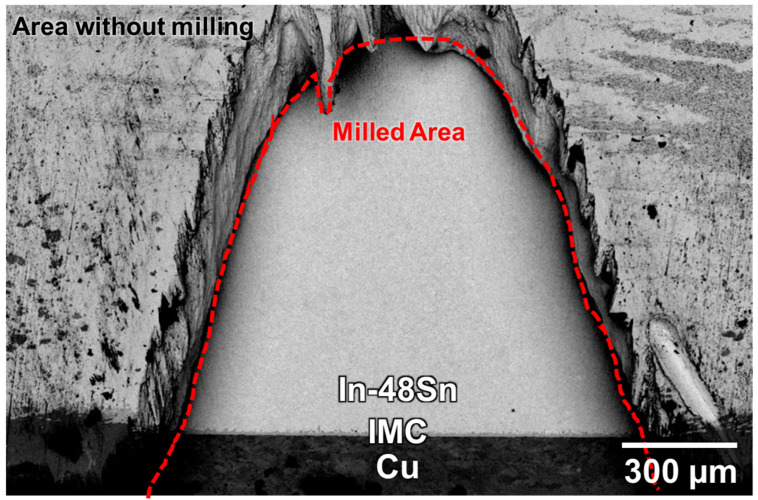
Cross-section micrograph showing the In-48Sn/Cu interface after ion milling.

**Figure 3 materials-16-03290-f003:**
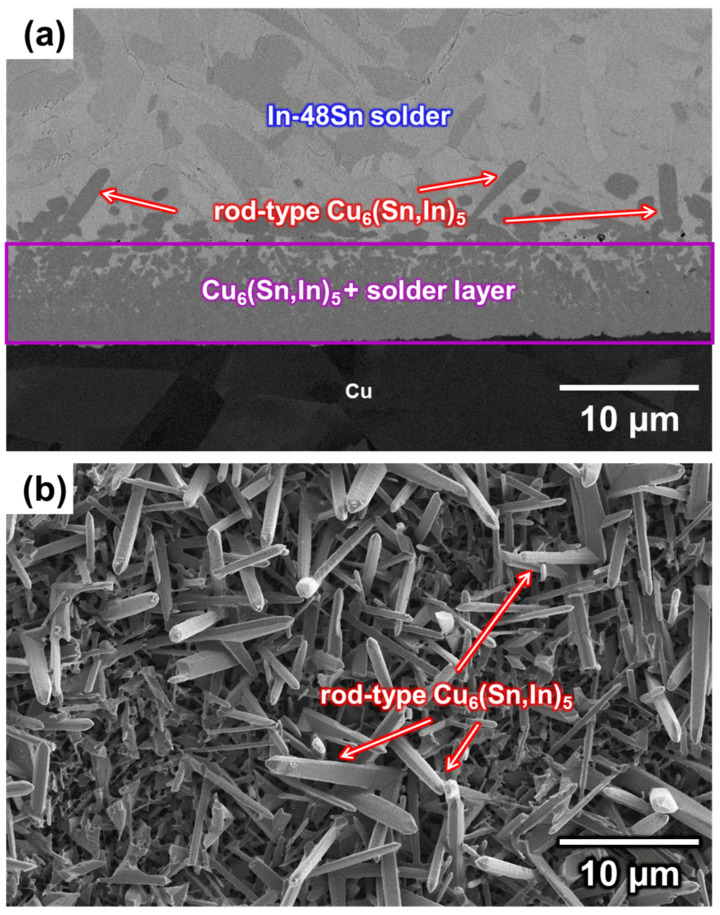
(**a**) Cross-section of the In-48Sn/Cu interface after soldering at 150 °C for 80 min and (**b**) top-view after removing all In-48Sn solder.

**Figure 4 materials-16-03290-f004:**
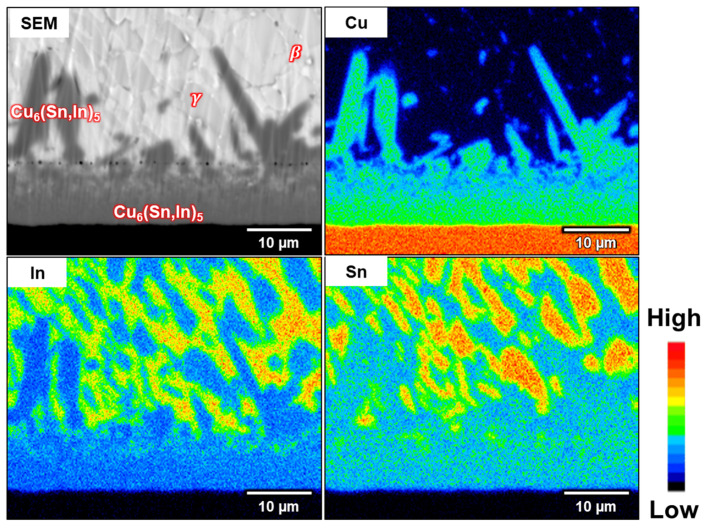
EPMA elemental mapping of the In-48Sn/Cu interface after soldering at 150 °C for 80 min.

**Figure 5 materials-16-03290-f005:**
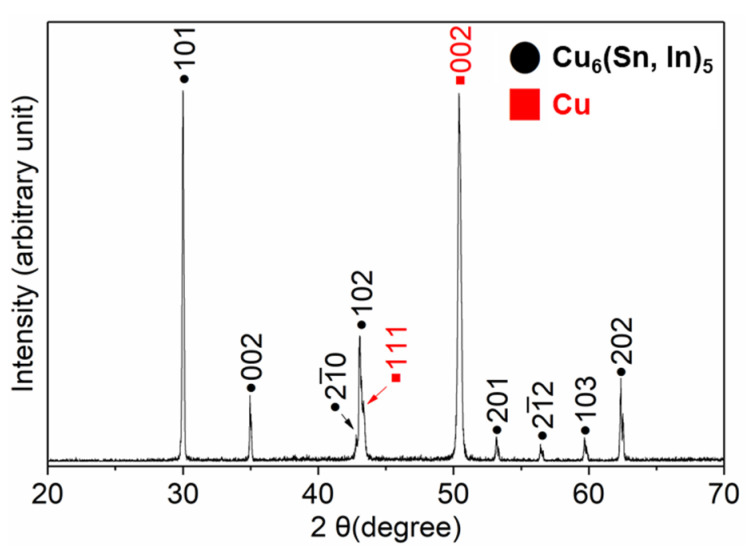
XRD pattern of Cu_6_(Sn,In)_5_ and Cu substrate after soldering at 150 °C for 80 min.

**Figure 6 materials-16-03290-f006:**
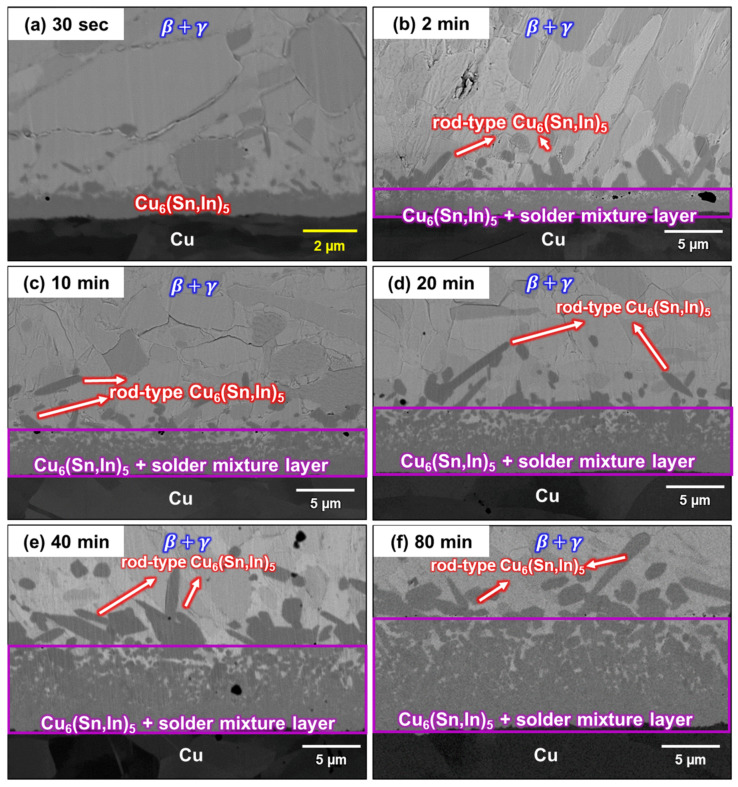
Micrographs of the In-48Sn/Cu interfaces after soldering at 150 °C for (**a**) 0.5 min, (**b**) 2 min, (**c**) 10 min, (**d**) 20 min, (**e**) 40 min, and (**f**) 80 min.

**Figure 7 materials-16-03290-f007:**
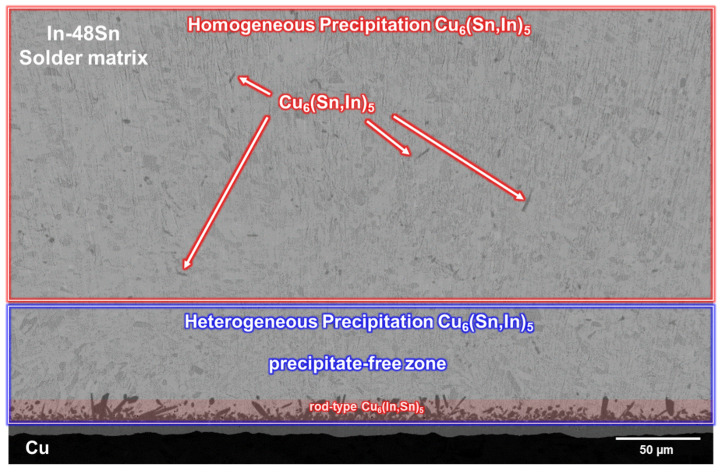
Micrograph of the In-48Sn/Cu interface after soldering at 150 °C for 80 min followed by water quenching.

**Figure 8 materials-16-03290-f008:**
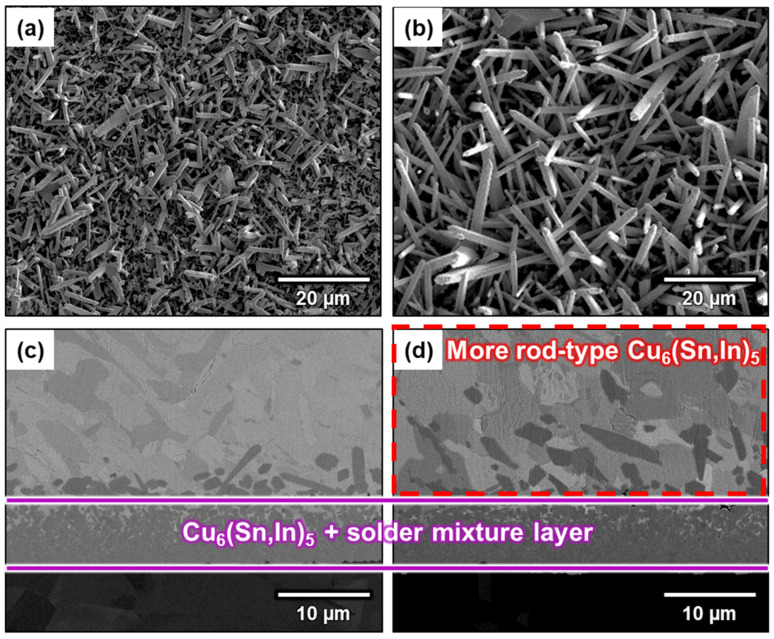
Micrographs of the interfaces after soldering at 150 °C for 80 min using different amounts of In-48Sn solder: (**a**) 0.15 g/top view, (**b**) 0.075 g/top view, (**c**) 0.15 g/cross-section, and (**d**) 0.075 g/cross-section.

**Figure 9 materials-16-03290-f009:**
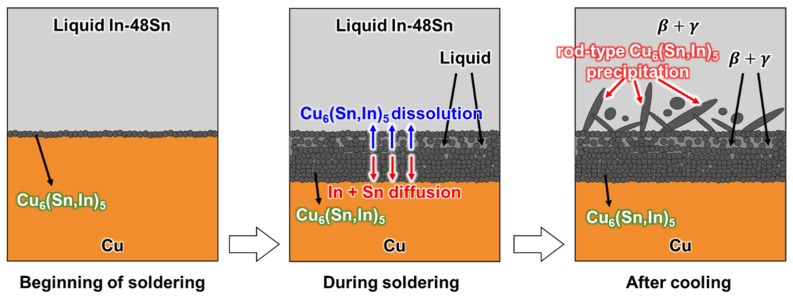
Schematic of the microstructure evolution between the liquid In-48Sn and solid Cu at 150 °C soldering.

**Figure 10 materials-16-03290-f010:**
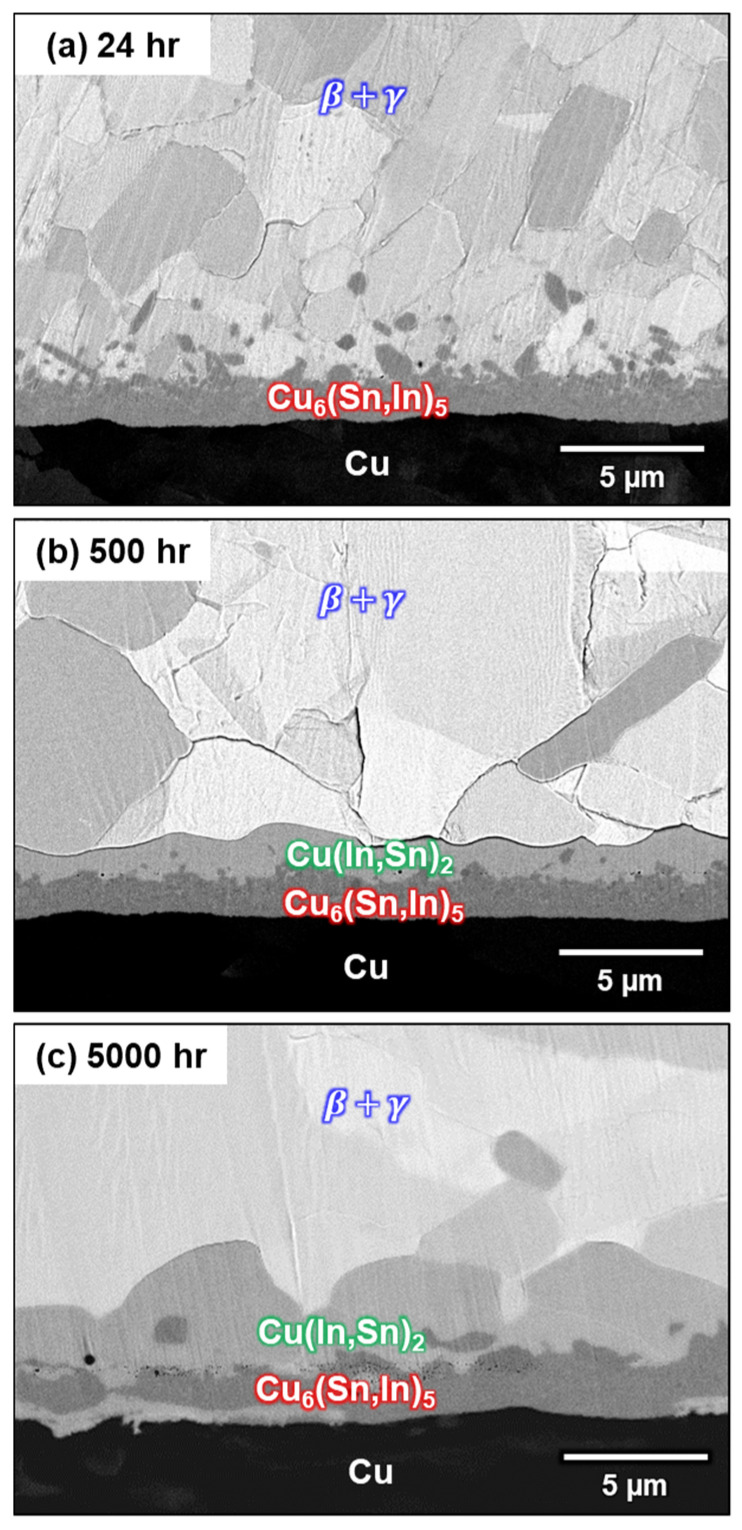
Micrographs of the In-48Sn/Cu interfaces after bonding and storage at 25 °C for (**a**) 24 h, (**b**) 500 h, and (**c**) 5000 h.

**Figure 11 materials-16-03290-f011:**
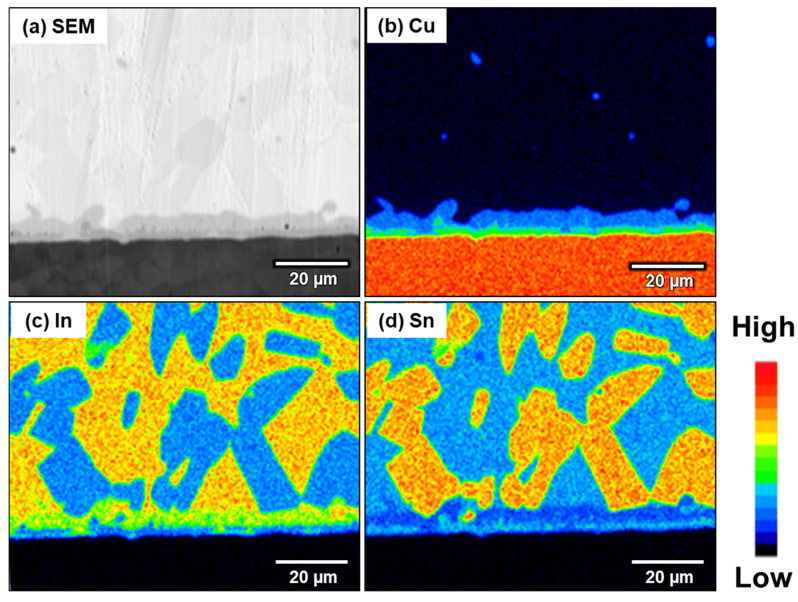
EPMA elemental mapping of the In-48Sn/Cu interface after room-temperature aging for 5000 h. (**a**) SEM, (**b**) Cu, (**c**) In, (**d**) Sn.

**Figure 12 materials-16-03290-f012:**
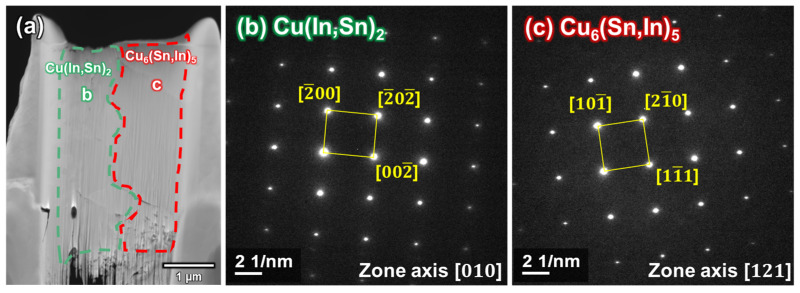
TEM imaging of In-48Sn/Cu interface after room temperature aging for 5000 h: (**a**) TEM bright field image of Cu(In,Sn)_2_ and Cu_6_(Sn,In)_5_, (**b**) SAED pattern of Cu(In,Sn)_2_, and (**c**) SAED pattern of Cu_6_(Sn,In)_5_.

**Figure 13 materials-16-03290-f013:**
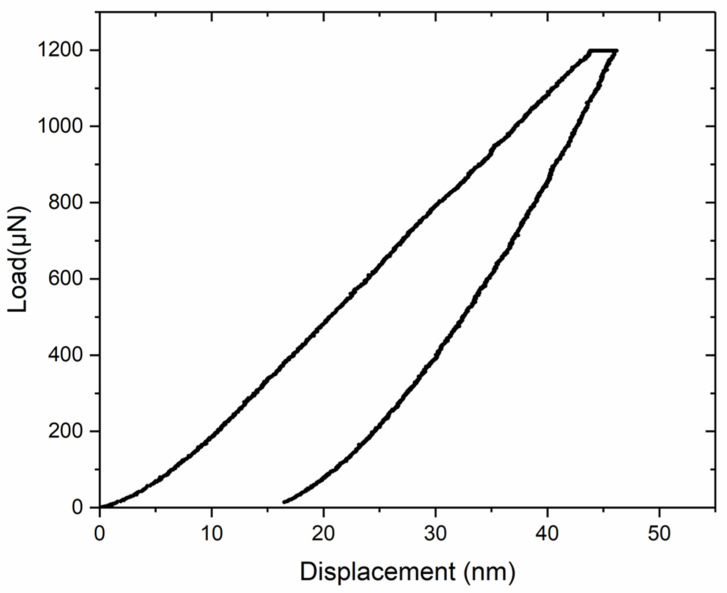
Load–displacement curves from the nanoindentation of Cu_6_(Sn,In)_5_.

**Table 1 materials-16-03290-t001:** EPMA quantitative analysis of *β*, γ, and Cu_6_(Sn,In)_5_.

Phase	Cu (at.%)	In (at.%)	Sn (at.%)
*β*	0.8	71.9	27.3
γ	2.2	26.1	71.7
Rod-type Cu_6_(Sn,In)_5_	55.7	18.9	25.4
Cu_6_(Sn,In)_5_ in the mixture layer	56.2	17.7	26.1

**Table 2 materials-16-03290-t002:** EPMA quantitative analysis of Cu(In,Sn)_2_ and Cu_6_(Sn,In)_5_.

Phase	Cu (at.%)	In (at.%)	Sn (at.%)
Cu(In,Sn)_2_	33.4	51.6	15.0
Cu_6_(Sn,In)_5_	57.3	17.3	25.4

**Table 3 materials-16-03290-t003:** Comparison of Young’s modulus and hardness between Cu_6_Sn_5_ and Cu_6_(Sn,In)_5_.

IMC Type	Young’s Modulus (GPa)	Hardness (GPa)
Cu_6_Sn_5_ [21]	118.97 ± 1.93	6.45 ± 0.14
Cu_6_(Sn,In)_5_	119.04 ± 3.94	6.28 ± 0.13

## Data Availability

Not applicable.

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
