# Peer review of "Artifact-Free Microstructures in the Interfacial Reaction between Eutectic In-48Sn and Cu Using Ion Milling"

_materials, 2023, doi:10.3390/ma16093290_

Round 1

Reviewer 1 Report

The article deals with a very relevant topic, namely lead-free solder In-48Sn and its features when soldering. The article is written competently, has scientific novelty and practical significance. It will be useful to other researchers. However, I have a few comments for the authors, which I hope will help the authors to improve the quality of the article and publish it in the scientific journal Materials.

1. In addition to the noted advantages of In-48Sn solder, one can add that it is very well processed by plastic deformation (https://doi.org/10.3390/machines9050093) and it is quite simple to obtain wire and rods of various diameters from it.

2. It is necessary to give a detailed description of the results of the studies made in articles [5-14] (lines 48-49). Indicate what exactly the authors investigated and what results they obtained.

3. The method for determining the mechanical properties of the Cu6(Sn,In)5 given in lines 257-259 should be moved to 2. Experimental method.

4. In Conclusions and Abstract, I recommend adding the numerical values of Young's modulus and the hardness of the Cu6(Sn,In)5 compound.

Author Response

Thank you for your comments, the point-by-point response is in the attached file.

Author Response

(The authors gave the same response as above.)

Reviewer 3 Report

The manuscript reports about IMCs phases and microstructure development of eutectic formulation by ion milling process. A sound methodology has been employed and the results are presented in a nice manner in imagery; however their presentation should be sufficiently improved in terms of text. Overall the article is good and can be published after addressing the following points.

1.      Title may be rechecked to include the process i.e. ion milling.

2.      Abstract should be improved by making it consisting of a quick introduction followed by a gap mentioning quick methodology and brief results. Gap-based objective should be an integral part at the start. The current version of abstract starts directly with the aim without any background or gap.

3.      Introduction section needs to be made comprehensive. Currently it seems to be an abridged version e.g. all the references 5-14 have been given simultaneously without addressing their individual jist.

4.      It is observed that the manuscript has largely been written in personal pronouns you and especially we. Good practice will rewrite such statements in passive voice to avoid personal pronouns.

5.      Similar to the above comment, the manuscript needs a re-check of grammatical mistakes as there are many such errors, which should be removed.

6.      Methodology section should also cover the methodological process of nano-indentation process.

7.      The authors should reflect on the effect of the obtained microstructure on the mechanical behavior. With comparison to the referenced properties, there is no difference in hardness and young’s modulus, while the better microstructure obtained should have sufficiently improved the mechanical behavior.

4.      It is observed that the manuscript has largely been written in personal pronouns you and especially we. Good practice will rewrite such statements in passive voice to avoid personal pronouns.

5.      Similar to the above comment, the manuscript needs a re-check of grammatical mistakes as there are many such errors, which should be removed.

Author Response

(The authors gave the same response as above.)

Round 2

Reviewer 2 Report

The authors improved the quality of the paper with the recommendations 

Reviewer 3 Report

Suggestions incorporated

minor check during typesetting required